# Cycloastragenol: A Novel Senolytic Agent That Induces Senescent Cell Apoptosis and Restores Physical Function in TBI-Aged Mice

**DOI:** 10.3390/ijms24076554

**Published:** 2023-03-31

**Authors:** Yanghuan Zhang, Dongxiao Gao, Yang Yuan, Runzi Zheng, Manting Sun, Shuting Jia, Jing Liu

**Affiliations:** Laboratory of Molecular Genetics of Aging and Tumor, Medical School, Kunming University of Science and Technology, Kunming 650500, China; zhangyh0701@163.com (Y.Z.); dongxiao0614@126.com (D.G.); yy1304504172@163.com (Y.Y.); 20212136032@stu.kust.edu.cn (R.Z.); mandyshuen@163.com (M.S.); shutingjia@kust.edu.cn (S.J.)

**Keywords:** CAG, senolytic, cell senescence, aging

## Abstract

Accumulating evidence indicates that the increased burden of senescent cells (SCs) in aged organisms plays an important role in many age-associated diseases. The pharmacological elimination of SCs with “senolytics” has been emerging as a new therapy for age-related diseases and extending the healthy lifespan. In the present study, we identified that cycloastragenol (CAG), a secondary metabolite isolated from *Astragalus membrananceus*, delays age-related symptoms in mice through its senolytic activity against SCs. By screening a series of compounds, we found that CAG selectively kills SCs by inducing SCs apoptosis and that this process is associated with the inhibition of Bcl-2 antiapoptotic family proteins and the PI3K/AKT/mTOR pathway. In addition, CAG treatment also suppressed the development of the senescence-associated secretory phenotype (SASP) in SCs, thereby inhibiting cell migration mediated by the SASP. Furthermore, the administration of CAG for 2 weeks to mice with irradiation-induced aging alleviated the burden of SCs and improved the animals’ age-related physical dysfunction. Overall, our studies demonstrate that CAG is a novel senolytic agent with in vivo activity that has the potential to be used in the treatment of age-related diseases.

## 1. Introduction

Aging is characterized by a decreased function of all tissues and an increased incidence of chronic diseases such as neurodegenerative pathologies, cardiovascular diseases, diabetes, and cancer [1]. More than 90% of people have at least one chronic age-associated disease by the age of 65 [2]. Thus, it is imperative to develop more efficient and cost-effective antiaging or healthy aging therapies. Several major factors have been identified as mechanisms of aging; these can be classified as the “nine hallmarks”, which include genomic instability, telomere attrition, mitochondrial dysfunction, and cellular senescence [3]. Among these hallmarks, senescent cells (SCs) accumulation is one of the causative processes of aging, and it is responsible for certain aging-related disorders [4]. Cellular senescence is a cell fate involving irreversible replicative arrest, apoptosis resistance, and increased production of inflammatory cytokines termed the senescence-associated secretory phenotype (SASP) [5]. There is accumulating evidence to demonstrate that SCs are involved in various age-related diseases, including type 2 diabetes, neurodegenerative disorders, cancer, atherosclerosis, and osteoarthritis [6]. A growing number of studies have demonstrated that the selective elimination of SCs in aged mice can be achieved via a transgene—a process known as senolysis—improving age-related deterioration of metabolic function and extending health span [7], as well as supporting the idea the SCs play a pivotal role in aging. Therefore, screening and identifying small molecules to specifically kill SCs or to inhibit the development of the SASP, termed senolytic agents, can extend the healthy and total lifespan.

Since the first report of senolytic drugs in 2015 [8], a handful of senolytic agents have been identified. Among these agents, the main targeting strategies are based on the characteristic SCs resistance to apoptosis that allows SCs survival [9,10]. In addition, a clinical trial successfully identified that a combination of quercetin and dasatinib (D + Q) induced SCs death and decreased the burden of SCs, as well as improving physical function in idiopathic pulmonary fibrosis [11]. However, most reported senolytics are dependent on cell type and exhibit substantial cytotoxicities in vivo, including neutropenia and platelet deficiency [12], limiting their potential use for clinical purposes. Thus, new senolytic agents are needed to eliminate SCs broadly and safely. It is worth noting that some natural compounds such as fisetin, piperlongumine, and curcumin analog EF24 also have senolytic activity in different senescent cell models [13,14,15]. The discovery of these natural senolytic compounds established the concept that the pharmacological elimination of SCs using natural senolytic compounds may have a better chance of being translated into clinical settings for the treatment of age-associated diseases, thanks to their advantage of low toxicity.

In this study, we screened the senolytic activities of a series of natural compounds reported to have antiaging effects [16]. Among these natural compounds, we identified cycloastragenol (CAG) (Figure 1A) as a novel senolytic agent which is able to specifically induce SCs death. CAG, a secondary metabolite isolated from *Astragalus membrananceus*, has been shown to have a broad range of pharmacological functions in preclinical models of disease, including telomerase activation, telomere elongation, anti-inflammatory effects, antioxidative properties, and improvement of lipid metabolism [17]. In addition, CAG is the only compound reported to activate telomerase in humans, and CAG can be relatively safely administered orally within a certain dose range [18]. Our previous study also demonstrated that CAG activates the circadian gene expression of SCs and restores the circadian rhythm in aging mice [19]. However, the senolytic activity and anti-aging action of CAG have not been clearly elucidated. To validate the screening platform, CAG was tested for senolytic activity in different human cell cultures and in a mouse aging model. Our results show that CAG can selectively decrease the viability of human SCs in a dose-dependent manner through the induction of apoptosis via inhibition of the antiapoptotic Bcl-2 family and the PI3K/AKT/mTOR pathway. Further analysis revealed that CAG also suppressed SASP expression and attenuated age-associated signatures in TBI-aged mice. These findings demonstrate that CAG is a potential novel senolytic agent which can be used to treat age-related diseases.

## 2. Results

### 2.1. CAG Is a Potential Senolytic Agent

To confirm senescence, SA-β-Gal activity was measured in human IMR-90 fibroblast nonsenescent cells (NCs) and etoposide-induced SCs (VP16-SCs) (Figure 1B). We then screened the senolytic activities of 10 natural compounds reported to have antiaging effects by measuring their effects on the viability of NCs and SCs (Appendix A). Cell viability assays showed that CAG dose-dependently reduced the viability of VP16-SCs but had no toxicity toward NCs (Figure 1C). To further confirm the senolytic activity of CAG, we used C_12_FDG to mark SCs and quantified the C_12_FDG-positive cells via flow analysis. As shown in Figure 1D, CAG reduced the number of SCs, consistent with senolytic activity. These results demonstrate that CAG is a potent and selective senolytic agent.

In addition, we assessed the viability of senescent HELF cells induced by VP16 (VP16-SCs) and extensive replication (Rep-SCs) after treatment with CAG. As shown in Figure 1E, CAG reduced the cell viability of both VP16-SCs and Rep-SCs, while it had a minimal effect on NCs viability. In addition, the expression levels of PAI-1 and P21, which are hallmarks of senescence, were significantly inhibited by CAG treatment (Figure 1F). These results demonstrate that CAG selectively eliminates SCs with different origins, suggesting CAG is a potential senolytic agent.

### 2.2. CAG Induces Senescent Cell Apoptosis by Inhibiting the Antiapoptotic Bcl-2 Family and PI3K/AKT/mTOR Pathway

Next, we investigated the mechanism by which CAG induces SCs death. First, we observed typical apoptotic morphological changes in SCs after CAG treatment, such as fragmented nuclei and apoptotic bodies (Figure 2A). Therefore, we hypothesized that CAG selectively kills SCs through the induction of apoptosis. Annexin V/propidium iodide (PI) double staining was used to detect apoptotic cells via flow cytometry after CAG treatment in SCs. As shown in Figure 2B, 100 μM CAG treatment increased the number of annexin-V-positive cells in SCs by 27.3%, compared with the DMSO group in which this number increased by 4%. These results suggest that CAG selectively killed SCs via induction of apoptosis.

A previous study reported that Bcl-2 and the cell proliferation pathway were activated in SCs to protect SCs from apoptosis and induce cell senescence [9,20]. To determine whether CAG induces SCs apoptosis via inhibition of the Bcl-2 family and PI3K/AKT/mTOR pathway, we examined the expression levels of related proteins using Western blots. The data showed that CAG significantly decreased the levels of Bcl-2 and PARP expression in senescent IMR-90 cells while reducing the levels of Bcl-xl and Bcl-2 in senescent HELF cells (Figure 2C). Furthermore, we found that CAG (100 μM) significantly reduced the levels of p-PI3K, p-AKT, and p-mTOR, suggesting that CAG functions in SCs by blocking the PI3K/AKT/mTOR pathway to inhibit cell senescence (Figure 2D). Taken together, these results suggest that CAG may selectively induce SCs apoptosis by inhibiting the Bcl-2 family and PI3K/AKT/mTOR pathway.

### 2.3. CAG Suppresses the SASP and Decreases Senescence and Cell Migration Induced by SCs

Given the efficacy of CAG in reducing SCs as a senolytic agent, we next investigated the potential of CAG to inhibit the SASP, which can mediate developmental senescence [5]. We found that CAG suppressed the mRNA levels of key SASP components, such as IL-6 and CXCL-10, in senescent IMR-90 cells. As the concentration of CAG increased, SASP mRNA levels were further reduced. Consistently, CAG also inhibited SASP expression in senescent HELF cells (Figure 3A). To further substantiate these data, we collected conditioned medium (CM) from SCs without or treated with CAG and measured the cytokine secretion using Western blot assays. The data showed that CAG treatment significantly reduced the secretion levels of MMP9, SDF1, and IL-6, which further confirmed the inhibition of the SASP by CAG (Figure 3B).

Previous studies have revealed functionally crucial factors such as p38 and JAK, which mainly regulate the SASP via NF-κB transcription [21]. To investigate the mechanisms by which CAG inhibits the SASP, we assessed the protein expression of the p38/NF-κB and JAK/STAT3 pathways. It was observed that CAG treatment significantly downregulated the expression levels of p-NF-κB and p-STAT3 (Figure 3C). These results suggest that CAG suppresses the SASP in SCs by inhibiting NF-κB and STAT3 activation.

Because the SASP can promote the senescence and migration of adjacent healthy cells [22], we tested whether SASP suppression by CAG could reduce the chemoattraction caused by SCs-CM. First, we cultured NCs using the CM of SCs which were pretreated with CAG. As shown in Figure 3D, the senescence markers, including PAI-1 and p21, were highly expressed in NCs treated with CM from SCs. These data suggest that the SASP could promote NCs senescence. However, the senescence markers were decreased when NCs were treated with the CM from SCs pretreated with CAG. Next, we investigated whether CAG could inhibit the cell migration induced by the SASP. The results from this study showed that cell migration was reduced in response to the CM from SCs pretreated with CAG. To rule out whether the inhibition of cell migration by CAG occurred in NCs, we also treated NCs with CM from SCs cultured in the presence of CAG (CAG was placed in the incubator for 24 h to mimic the CM collection process). Our data show that CAG had little effect on reducing cell migration induced by CM from SCs (Figure 3E). Therefore, CAG may inhibit cell migration by suppressing the SASP in SCs rather than by acting on NCs to decrease SASP development. Together, these findings indicate that CAG plays a role in regulating the SASP.

### 2.4. CAG Can Synergistically Eliminate SCs with ABT263

ABT263 is a potent senolytic agent that can induce SCs apoptosis via inhibition of the Bcl-2 family [9]. However, ABT263 also induces platelet apoptosis and results in severe thrombocytopenia, which largely limits the use of ABT263 in clinical settings. Therefore, we wondered whether CAG and ABT263 could synergistically eliminate SCs, which could lower the required dose of ABT263 and thus reduce its toxicity. We measured the cell viability of NCs and SCs after treatment with ABT263 and CAG. Interestingly, ABT263 showed a significant inhibition effect on the cell viability of NCs, which could be attenuated by CAG. However, the combination of ABT263 and CAG had a better inhibitory effect on the cell activity of SCs than individual treatment (Figure 3F), indicating that the drug combination exerted a strong senolytic effect on SCs.

### 2.5. CAG Delays Age-Related Symptoms and Organ Aging in TBI-Aged Mice

Given the effect of CAG on eliminating SCs in vitro, we next investigated whether CAG could protect against age-related pathologies in vivo. First, we induced cell senescence by exposing C57BL/6 mice to total-body irradiation (TBI) at a sublethal dose (5 Gy). Two months after TBI, when the irradiated mice had developed an age-related phenotype, including grayed fur and rickets, we treated mice with 50 mg/kg CAG or vehicle only for 2 weeks by oral gavage (Figure 4A). The body weight was measured every 3 days, and the mice were sacrificed 1 week after the last dose. Consistent with the results of our previous study [19], the administration of 50 mg/kg CAG daily had no substantial effect on the body weight (Figure 4B) and organ coefficients (Figure 4C), suggesting that CAG is available as an oral dietary supplement and has no adverse side-effects. It was noted that the abnormal body appearance, including markedly grayed fur and rickets, was largely reversed by CAG treatment (Figure 4D). A number of studies have reported that aged mice exhibit declines in behavioral performance; thus, we evaluated the locomotor activity and behavioral responses of the mice using an open-field test. As expected, TBI significantly decreased the total distance traveled, vertical activity, and time spent in the center area. By contrast, CAG administration provided a substantial benefit in restoring these activities (Figure 4E).

To further examine the effect of CAG on aging organs, we selected the small intestine and bone, which exhibit high sensitivity to TBI-induced aging [23,24]. As shown in Figure 4F, TBI induced a decrease in the number of GLR5-positive cells in the intestinal crypts, suggesting radiation-induced exhaustion of stem cells in the intestinal crypts. However, CAG treatment increased the expression of GLR5 in the intestinal crypts compared with vehicle treatment. In addition, micro-CT analysis showed that TBI also induced 3D bone mineral density (BMD) decrease in the femur, which was reversed by CAG treatment. The total average bone mineral density was 431.6 mg/cm^3^ in the CAG group and 394.5 mg/cm^3^ in the TBI group (Figure 4G). Taken together, these data demonstrate that CAG improved the age-related symptoms and organ aging in TBI-aged mice.

### 2.6. CAG Decreases the Burden of Senescent Cells and Reduces SASP in TBI-Aged Mice

To determine whether the delay of age-related symptoms in CAG-treated mice coincided with a reduction in the number of SCs in tissues, we first collected samples of inguinal adipose tissue (IAT) and analyzed the number of SCs using SA-β-gal staining. As shown in Figure 5A, the IAT of TBI-aged mice stained strongly for SA-β-Gal, while CAG-treated mice showed a marked decrease in SA-β-Gal staining. In addition, senescence markers including P53, P21, and P16 were significantly elevated in the liver of TBI-aged mice compared to age-matched control mice. Treatment with CAG resulted in a significant reduction in the protein expression levels of senescence markers, as well as in the mRNA expression of *p16* (Figure 5B,C). Similarly, CAG treatment also decreased the mRNA expression of SASP factors including *IL-6*, *CXCL-5*, and *CXCL-10* in liver tissues compared with the vehicle group (Figure 5D).

We next investigated whether the elimination of senescent cells and SASP reduction in TBI-aged mice was consistent with the signal pathways in vitro. As shown in Figure 5E, the expression levels of BCL2, p-AKT, p-mTOR, and p-STAT3 were significantly increased in TBI-aged mice. Notably, the expression levels of these proteins were significantly reduced in liver tissues compared to the vehicle-treated group, a pattern consistent with the in vitro data. Taken together, these data demonstrate that CAG reduced the burden of SCs and decreased the SASP in aged mice.

## 3. Discussion

Aging is a complex and inevitable process that often results in physiological degeneration. The chronic accumulation of SCs has emerged as a hallmark and driver of age-related phenotypes [25]. Interestingly, the deletion of senescent cells through a genetic or pharmacological approach attenuated age-related tissue deterioration and extended the healthy lifespan and total lifespan in mice [26]. Hence, this seminal finding prompted research to identify small molecules that can selectively eliminate SCs or suppress SASP production, termed senolytics or senomorphics, which offer potential as a therapeutic strategy for age-related diseases. To date, several classes of senolytics have been reported, including dasatinib, quercetin, fisetin, HSP-90 inhibitors, and ABT-263 (Bcl-2 family inhibitors) [27]. However, most of the reported senolytics have various toxicities, such as thrombocytopenia and hepatotoxicity [28], and some senolytics need further study in vivo. Therefore, there is a need to identify new compounds with broad-spectrum senolytic activity and low cytotoxicity.

In this study, we screened a series of compounds reported to have antiaging effects with the aim of identifying new senolytic agents with efficacy and safety both in vitro and in vivo (Figure 6). As a result, we identified CAG, a secondary metabolite isolated from *Astragalus membrananceus*, as a new, natural, and potent senolytic, which selectively induces cell death in SCs but not NCs. It is noteworthy that CAG also suppresses SASP expression, meaning it can act as a senomorphic to reduce the impact of SCs on the age-related phenotype. Increasing evidence shows that CAG has a wide spectrum of pharmacological functions, including telomere elongation, telomerase activation, anti-inflammatory effects, and antioxidative properties [17]. However, the underlying mechanisms associated with CAG are not clear; thus, we further investigated the potential senolytic activity and mechanisms of CAG.

Apoptosis is a highly complex physiological process that is particularly mediated by a family of proteins called the Bcl-2 family. The activation of proapoptotic members of the Bcl-2 family leads to the release of proapoptotic proteins, such as cytochrome c, from the mitochondrion. Then, the apoptosome activates caspase-9, which leads to the activation of caspase-3 and results in apoptosis [29]. Previous studies have found that SCs are characterized by resistance to apoptosis via upregulation of the Bcl-2 family of antiapoptotic proteins, including Bcl-xl and Bcl-2 [30]. To date, several classes of senolytics, such as ABT263 and EF24, have been developed to selectively kill SCs via targeted inhibition of these proteins [27]. Consistent with these previous findings, we observed that CAG treatment of SCs induced apoptosis in a dose-dependent manner. Further analysis revealed that CAG treatment significantly reduced the expression levels of Bcl-xl and Bcl-2 while enhancing the cleaved PARP level, further confirming induction of the apoptosis program in SCs. It appears that CAG may promote proteasome degradation of the Bcl-2 family in SCs. In addition, we found that CAG can synergistically eliminate SCs when combined with ABT263. Drug combinations to reduce the thrombocytopenia associated with ABT263 in vivo are of great interest. It has been widely confirmed that activation of the AKT/mTOR signaling pathway induces age-related accumulation of DNA damage and results in cell senescence [31]. We next hypothesized that the AKT/mTOR pathway could be involved in regulating the senolytic effect of CAG. We observed a dose-dependent inhibition of the phosphorylation of PI3K, AKT, and mTOR after CAG treatment. Taken together, it is reasonable to conclude that CAG-mediated inhibition of the Bcl-2 family of antiapoptotic proteins and the AKT/mTOR pathway may have contributed to SCs-induced apoptosis, which caused a decrease in cell viability.

Since many of the negative effects associated with senescence are driven by the SASP, we further investigated whether the senolytic property of CAG could modulate the senescence secretome. We found that CAG suppressed particular SASP components and reduced the inflammation and chemoattraction caused by SCs. Further analysis revealed that p-STAT3 and p-NF-κB were significantly reduced by CAG. In fact, bioinformatic analysis of several aged tissues demonstrated that various cytokines related to JAK/STAT signaling were upregulated in senescent cells, while JAK inhibitors suppressed the SASP and alleviated frailty in aged mice [32,33]. Furthermore, several studies have shown that NF-κB is a main transcription factor enabling the activation of the SASP in response to senescence inducers, and that inhibition of NF-κB activation in SCs by metformin can selectively represses SASP genes that require this transcription factor [34,35]. Thus, CAG can ultimately aid in suppressing the escalation of senescence to neighboring cells, suggesting a dual mechanism of this natural agent in targeting SCs, similar to that of the natural product PCC1 [36]. However, more work is needed to elucidate the potential senomorphic effects of CAG in different SCs.

Through existing studies and clinical trials, CAG has been identified as an antiaging compound with extensive pharmacological functions, including telomere elongation, telomerase activation, anti-inflammatory effects, antioxidative properties, and improvement of lipid metabolism [17]. Here, we showed for the first time a similar antiaging effect of CAG on physical function and SCs in multiple tissues of TBI-aged mice. First, we confirmed that prolonged administration of CAG is relatively safe within a certain dose range. Second, CAG significantly delayed the onset of multiple age-related symptoms in TBI-aged mice, including grayed fur, rickets, osteoporosis, and a decline in behavioral performance. Third, CAG exerted an elimination effect on SCs in the small intestine and IAT, as well as decreasing the expression of senescence markers and SASP-associated genes in aged liver. Most importantly, the expression levels of apoptosis resistance and inflammation-related proteins in the CAG-treated group were also significantly inhibited compared with vehicle-treated aged mice, consistent with the in vitro data. Given that CAG is a safe natural product with broad application prospects, our preclinical study suggests that CAG may be a novel senolytic agent that is imminently applicable with benefits for health and longevity.

## 4. Material and Methods

### 4.1. Cell Culture, SC Induction, and Reagents

Human IMR-90 fibroblasts were purchased from the American Type Culture Collection (ATCC, Manassas, VA, USA). Human HELF fibroblasts were obtained from the Kunming Institute of Zoology. Cells were cultured in complete Dulbecco’s modified Eagle’s medium (DMEM, Cat. C11995500BT, Thermo Fisher Scientific, Waltham, MA, USA) supplemented with 10% heat-inactivated fetal bovine serum (FBS, Cat. 16000044, Thermo Fisher Scientific, Waltham, MA, USA) in a humidified incubator at 37 °C and 5% CO_2_.

Replicative exhaustion and DNA damage were used for the induction of SCs, as previously described [37]. Briefly, low-passage IMR-90 and HELF cells (<25 passage) were used as nonsenescent cells (NCs). To induce replicative senescence (Rep-SCs), IMR-90 and HELF were cultured until they stopped dividing and became growth-arrested after 38 passages. To induce SCs via DNA damage, cells were treated with 20 μM etoposide (VP16-SCs) for 24 h. Forty-eight hours after etoposide removal, >70% of the cells were SA-β-gal^+^. CAG (Cat. No. HY-N1485) and other natural compounds were purchased from MedChemExpress (MCE, NJ 08852, USA).

### 4.2. SA-β-Galactosidase Staining of Cultured Cells and Adipose Tissue

SCs were detected by measuring β-galactosidase activity according to the manufacturer’s procedure (Cat. C0602, Beyotime Institute of Biotechnology, Haimen, China). In brief, after inducing senescence, cells were washed with PBS and fixed with stationary liquid for 15 min at room temperature. The cells were washed three times and stained with β-galactosidase staining solution, before being left overnight in the dark at 37 °C. On the next day, the cells were observed under an optical microscope, with blue color indicating the β-galactosidase activity of SCs. We stained adipose tissue depots for SA-β-Gal activity as previously described [38].

### 4.3. Assay to Identify Senolytic Agents

To identify the senolytic activities of natural compounds, the cell viability of NCs and SCs was assessed using the 3-(4,5-dimethyl-2-thiazolyl)-2,5-diphenyl-2*H*-tetrazolium bromide (MTT, M2128, Sigma Aldrich, St. Louis, MO, USA) assay as previously described [39]. Briefly, cells were treated with 50 and 100 μM natural compounds for 48 h, followed by the addition of MTT (5 mg/mL), and then further incubated for 4 h at 37 °C in a humidified CO_2_ incubator. DMSO (150 μL/well, Cat. 276855, Sigma Aldrich, St. Louis, MO, USA) was used to dissolve formazan crystals, and the absorbance at 490 nm was measured using a microplate reader (Bio-Tek Inc., Winooski, VT, USA).

To further confirm the senolytic activity of CAG, the SA-β-Gal activity was measured via 5-dodecanoylaminofluorescein-dib-D-galactopyranoside (C_12_FDG, Invitrogen, Waltham, MA, USA, Cat. I-2904) using flow cytometry, as previously described [10]. Briefly, lysosomal alkalinization was induced by pretreating cells with 100 nM bafilomycin A1 for 1 h. Then, cells were digested into single cells via trypsinization and incubated with 33 μM C_12_FDG for 2 h. The cells were washed with PBS after staining and analyzed using flow cytometry (Accuri C6, BD Biosciences, San Jose, CA, USA).

### 4.4. Apoptosis Assay

SCs were treated with vehicle or CAG (50 and 100 μM) for 48 h. The suspended cells were first collected and then pooled with the adherent cells detached using 0.1% trypsin. The cells were stained using an annexin V/fluorescein isothiocyanate (FITC) and PI apoptosis kit according to the manufacturer’s instructions (Cat. 11988549001, Roche, Basel, Switzerland). The stained cells were analyzed using flow cytometry (Accuri C6, BD Biosciences, San Jose, CA) within 1 h.

### 4.5. Western Immunoblotting

Cells or tissues were homogenized in RIPA lysate to extract total protein. Total protein contents were denatured by boiling at 95 °C for 10 min in a sample buffer (Bio-Rad Laboratories, Inc., Hercules, CA, USA). Proteins were separated using SDS-PAGE electrophoresis and transferred to a PVDF membrane (Cat. 1620256, Bio-Rad, Hercules, CA, USA). After blocking at room temperature for 1 h using PBST containing 5% nonfat dry milk, the membranes were incubated with primary antibodies overnight at 4 °C with gentle shaking. Signals were developed with HRP-conjugated secondary antibodies (Santa Cruz Biotechnology, Inc., Dallas, TX, USA) and chemiluminescent HRP substrate (Pierce). The blotting membranes were recorded using an enhanced chemiluminescence (ECL, 180-5001, Tanon, Shanghai, China) detection system (GE Healthcare, Piscataway, NJ, USA). All of the primary antibodies were purchased from Cell Signaling Technology, Inc. (Danvers, MA, USA).

### 4.6. Quantitative Polymerase Chain Reaction (qRT-PCR)

Total RNA was extracted from cells or tissues using TRIZOL reagent (Cat. 15596081, Life Technologies, Carlsbad, CA, USA), and 1 μg of RNA was reverse-transcribed to cDNA using an RNA Reverse Transcription Kit (Cat. LS2050, Promega, Madison, WI, USA). qRT-PCR was performed in a Real-Time PCR System 7300 using SYBR Mix (Cat. 04913914001, Roche, Basel, Switzerland), following a program consisting of 95 °C for 10 min, then 40 cycles of 95 °C for 15 s and 60 °C for 1 min. The primers for specific human and mouse genes were shown separately in Table 1 and Table 2. Data analysis was performed using the 2^(−△△Ct)^ method.

### 4.7. Cell Migration Assay

Migration assays were performed using transwell polycarbonate membrane inserts with a 5 μm pore diameter (Cat. 3378, Corning, Corning, NY, USA). Conditioned medium (CM) collected from SCs was introduced into the bottom chambers of the plates in triplicate. HELF NCs were plated into the inserts and allowed to migrate into the bottom chamber for 6 h at 37 °C. Then, the number of cells that migrated into each lower chamber was quantified using the crystal violet staining assay (Life Technologies).

### 4.8. Total-Body Irradiation (TBI) and CAG Treatment

C57BL/6 mice were obtained from the Nanjing Model Animal Center and maintained under specific pathogen-free facility (SPF) conditions with a 12 h light/dark cycle and free access to food and water. To induce senescence in vivo, male mice at 8–12 weeks of age were exposed to a sublethal dose (5 Gy) of TBI. Two months later, animals were randomly assigned to control or treatment groups and received vehicle or CAG (prepared in 5% CMC-Na, 50 mg/kg) via daily oral administration for 2 weeks. All animal experiments were performed according to the Animal Protection Guidelines of Kunming University of Science and Technology, China SYXK (Dian) K2018-0008.

### 4.9. Open-Field Test

The locomotor activity and behavioral responses of mice were evaluated using the open-field test. This test uses a camera to measure the movement of a test animal in the peripheral and central zones of a polyvinyl chloride box (O’Hara and Co., Ltd., Tokyo, Japan). In this test, a mouse was placed in a corner of the open-field apparatus. The total distance traveled (cm), line crossing numbers (measured by counting the number of photobeam interruptions), and time spent in the center area (20 × 20 cm) were recorded for 10 min using a video-imaging system (EthoVisionXT; Noldus Information Technology, Wageningen, The Netherlands), as described previously [40]. Each mouse was tested separately.

### 4.10. Micro-CT Analysis

The bone mineral density (BMD) of the femurs from TBI or sham control mice was determined after dissection, cleaning by removing adherent tissues, fixing in 4% PFA for 12 h, and storage in 100% ethanol. Bones were scanned using micro-CT (Inveon, Siemens, Erlangen, Germany) at a spatial resolution of 55 kVp, 145 mA, 300 ms integration time, and a voxel resolution of 20 µm obtained from 720 views [41].

### 4.11. Immunohistochemistry Staining

Mouse tissues (small intestine and liver) were collected at necropsy, placed in 10% neutral buffered formalin for 24 h, and then transferred to 70% alcohol. Then, the tissues were embedded in paraffin blocks for sectioning for immunohistochemistry staining as previously described [42]. For immunohistochemistry, the endogenous peroxidases of tissue slides were quenched in 0.3% H_2_O_2_ for 10 min and then blocked with 10% goat serum. The primary antibody in blocking buffer (1:50) was applied to tissues overnight at 4 °C. Sections were washed in PBS and then incubated with goat anti-rabbit (Cat. K5007, Dako, Glostrup, Denmark) or mouse IgG/horseradish peroxidase (BD Biosciences, San Jose, CA, USA) according to the standard protocols provided by the manufacturer, followed by incubation of Vectastain ABC Kit (Vector Laboratories, Newark, CA, USA). The positive area and integral optical density were examined under an inverted microscope at 200× magnification (Eclipse TS100, Nikon, Tokyo, Japan).

### 4.12. Statistical Analyses

Data were analyzed and expressed as mean ± standard deviation (SD). The data were analyzed using GraphPad Prism 8.0 (GraphPad Software, La Jolla, CA, USA). The comparisons were evaluated using the two-tailed Student’s *t*-test, analysis of variance (ANOVA), and Tukey’s multiple-comparisons test. A *p-*value < 0.05 was considered to be significant.

## Figures and Tables

**Figure 1 ijms-24-06554-f001:**
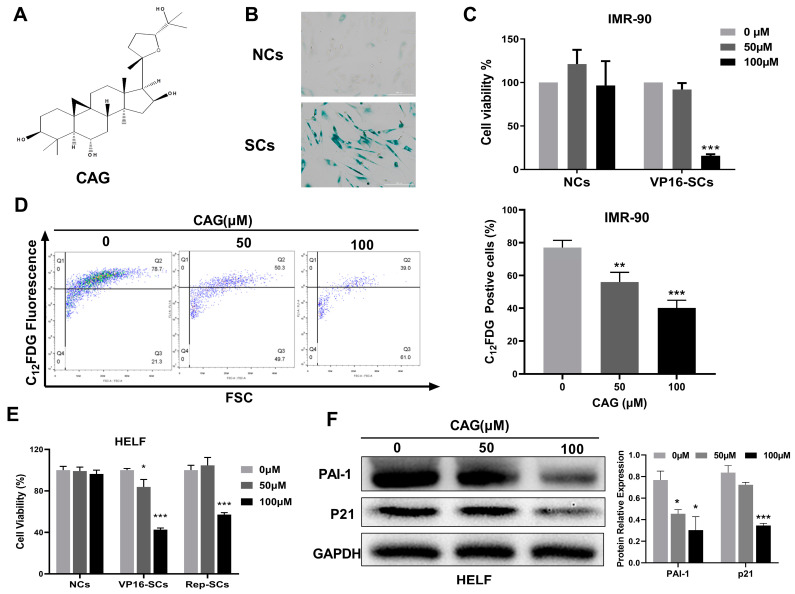
CAG is a potential broad-spectrum senolytic agent. (**A**) Chemical structure of CAG. (**B**) Representative images displaying SA-β-Gal staining after treatment of IMR-90 cells with 20 μM VP-16. Microscopic (magnification of 200×) images were taken. (**C**) Survival analysis of nonsenescent (NCs) and senescent IMR-90 cells (VP16-SCs) treated with CAG (at concentrations of 50 and 100 μM, respectively). (**D**) Representative flow cytometric histogram showing detection of SA-β-Gal activity using C12FDG in SCs after CAG treatment. The line indicates the cut-off in intensity levels used to define SCs (the cells above the line are SCs). (**E**) Survival analysis of nonsenescent (NCs) and senescent HELF cells (VP16-SCs and Rep-SCs) treated with CAG (at concentrations of 50 and 100 μM, respectively). (**F**) The expression of PAI-1 and P21 in senescent HELF cells after incubation with indicated concentrations of CAG. Protein levels were determined using Western blots. The data were derived from three biological replicates. * *p* < 0.05, ** *p* < 0.01, and *** *p* < 0.001.

**Figure 2 ijms-24-06554-f002:**
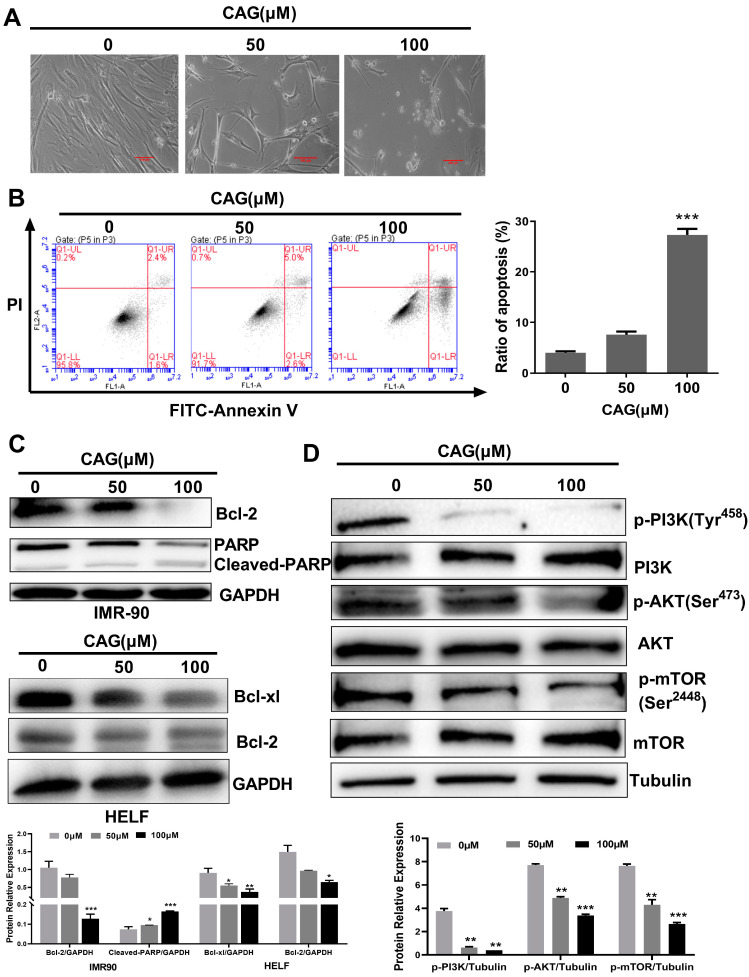
CAG induces SCs apoptosis by inhibiting the antiapoptotic Bcl-2 family and PI3K/AKT/mTOR pathway. (**A**) Cell morphological changes in SCs after CAG treatment. Microscopic (magnification of 200×) images were taken. (**B**) Flow cytometric analysis of the cell death of SCs treated with CAG via annexin V/PI staining. (**C**,**D**) The expression levels of Bcl-2, PARP, Bcl-xl, and PI3K/AKT/mTOR pathway in SCs after incubation with indicated concentrations of CAG. Protein levels were determined using Western blots. The value represents the protein expressions compared to the GAPDH or Tubulin. * *p* < 0.05, ** *p* < 0.01, and *** *p* < 0.001.

**Figure 3 ijms-24-06554-f003:**
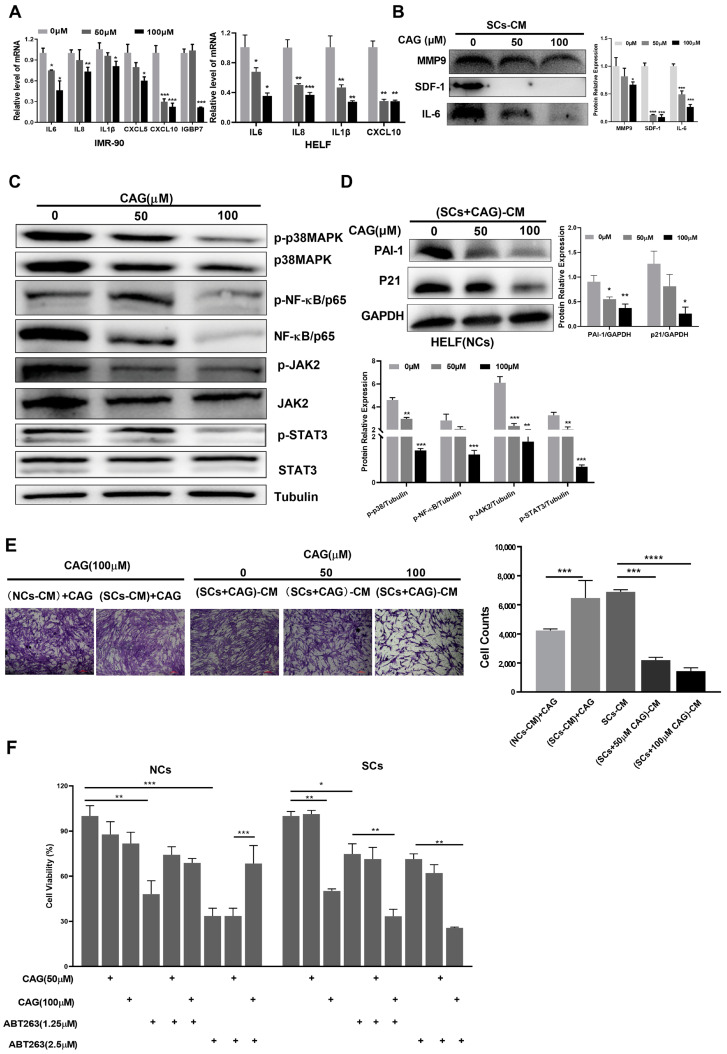
CAG suppresses the SASP and decreases senescence and cell migration induced by SCs. (**A**) The relative mRNA abundance of key SASP components in senescent IMR-90 and HELF cells treated with different concentrations of CAG. RNA was collected and real-time PCR was performed. (**B**) CM was collected from senescent HELF cells (SCs-CM). Cytokine protein levels in CM were measured using Western blots. (**C**) The expression levels of p38, MAPK, NF-κB, JAK, and STAT3 in SCs after incubation with indicated concentrations of CAG. Protein levels were determined using Western blots. (**D**) NCs were treated for 48 h with CM collected from SCs which was pretreated with CAG. Expression levels of PAI-1 and P21 were determined using Western blots. (**E**) HELF cells were placed in inserts in the top chambers, and CM from different groups (SC-CM: CM collected from senescent HELF cells; (SC-CM) + CAG: CM collected from senescent HELF cells with added CAG; (SC + CAG)-CM: CM collected from senescent HELF cells which was pretreated with CAG) was introduced into the lower chambers. Cell migration was tested using crystal violet staining following 24 h of exposure. Microscopic (magnification of 200×) images were taken. Data are shown as the fold-change in cell number in each treatment vs. the SC-CM group. (**F**) HELF-VP16-induced senescent cells were treated with indicated concentrations of CAG and ABT263 for 48 h. Cell viability was assayed via MTT assay. * *p* < 0.05, ** *p* < 0.01, *** *p* < 0.001 and **** *p* < 0.0001.

**Figure 4 ijms-24-06554-f004:**
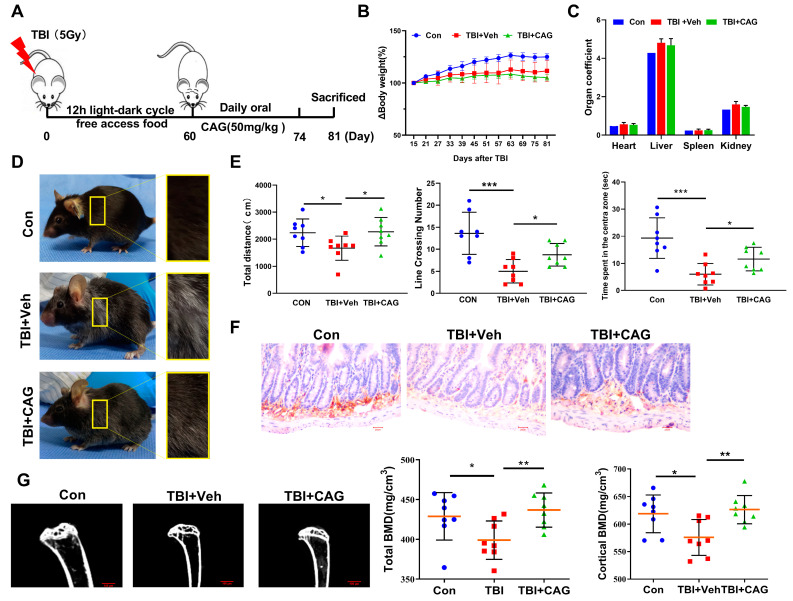
CAG delays age-related symptoms and organ aging in TBI-aged mice. (**A**) Schematic representation of the scheduling of TBI exposures. When the irradiated mice developed an age-related phenotype (about 2 months after TBI), the mice were treated with 50 mg/kg CAG or vehicle only for 2 weeks by oral gavage (*n* = 8 animals/group). (**B**,**C**) The body weight and organ coefficients of mice in different groups were measured. (**D**) Whole-body snapshot comparison of control mice exposed to TBI followed by vehicle treatment, and mice exposed to TBI and treated with CAG. (**E**) The open-field test including total distance traveled (cm), line crossing number, and center time (s) were recorded for each 5 min block of testing. (**F**) Representative images of GLR5-positive cells in the small intestinal crypts with immunohistochemistry staining. Microscopic (magnification of 400×) images were taken. (**G**) Representative 3D trabecular bone images of femur and quantification of bone mineral density (BMD) in different mouse groups using micro-CT after sacrifice. * *p* < 0.05, ** *p* < 0.01, and *** *p* < 0.001.

**Figure 5 ijms-24-06554-f005:**
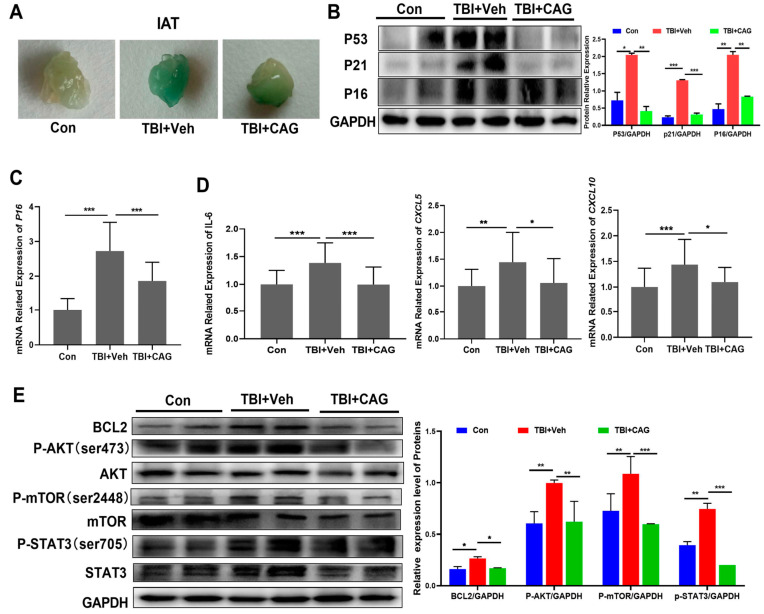
CAG decreases the senescent cell burden and reduces SASP in TBI-aged mice. (**A**) Representative images of SA-β-Gal staining of IAT from young and TBI-aged mice treated with vehicle or CAG. (**B**,**E**) The protein expression levels of senescence markers (**B**) and apoptosis pathway proteins (**E**) in liver from young and TBI-aged mice treated with vehicle or CAG. (**C**,**D**) The mRNA expression levels of *p16* and SASP in liver from young and TBI-aged mice treated with vehicle or CAG. * *p* < 0.05, ** *p* < 0.01, and *** *p* < 0.001.

**Figure 6 ijms-24-06554-f006:**
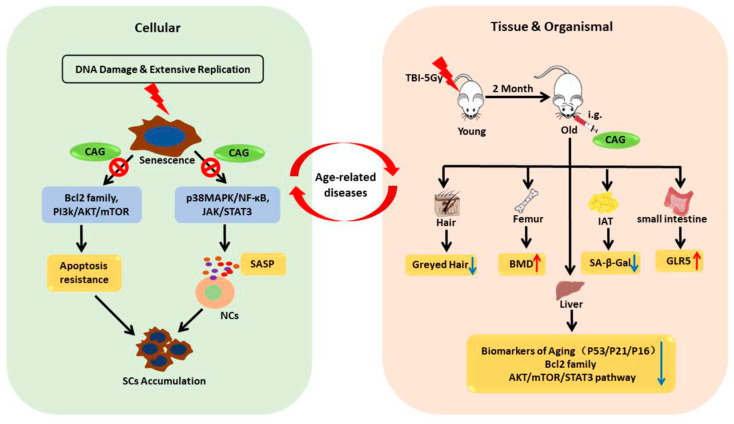
Illustration summarizing the key points of our findings. The accumulation of senescent cells is one of the causative processes of aging, while the aging of tissues and organs also promotes the accumulation of SCs. CAG selectively induces SCs apoptosis via inhibition of Bcl-2 and the PI3K/AKT/mTOR pathway. CAG also suppresses the development of the SASP in SCs, thereby inhibiting cell senescence which is mediated by the SASP. CAG alleviated the SCs burden and improved age-related physical dysfunction in TBI-aged mice. Therefore, the senolytic agent CAG has the potential to be used in the treatment of age-related diseases. (CAG: cycloastragenol; NC: nonsenescent cells; SCs: senescent cells; SASP: senescence-associated secretory phenotype, e.g.,: intragastrical administration; TBI: total-body irradiation; IAT: inguinal adipose tissue).

**Table 1 ijms-24-06554-t001:** Primers for specific human genes.

Gene	Forward Primer 5′ to 3′	Reverse Primer 5′ to 3′
*IL6*	TCTATACCACTTCACAAGTCGGAG	AGAATTGCCATTGCACAACTCTTT
*IL8*	ACTGAGAGTGATTGAGAGTGGAC	AACCCTCTGCACCCAGTTTTC
*IL1β*	ATGATGGCTTATTACAGTGGCAA	GTCGGAGATTCGTAGCTGGA
*CXCL5*	AGCTGCGTTGCGTTTGTTTAC	TGGCGAACACTTGCAGATTAC
*CXCL10*	AGAACGGTGCGCTGCAC	CCTATGGCCCTGGGTCTC
*IGFBP7*	CGAGCAAGGTCCTTCCATAGT	GGTGTCGGGATTCCGATGAC
*GAPDH*	GTCTTCACCACCATGGAGAAGGC	TTGTTGTCATGGATGACCTTGGCC

**Table 2 ijms-24-06554-t002:** Primers for specific mouse genes.

Gene	Forward Primer 5′ to 3′	Reverse Primer 5′ to 3′
*p16*	TACCCCGATTCAGGTGAT	TTGAGCAGAAGAGCTGCTACGT
*IL6*	GCCATCTTTGGAAGGTTCAGGTTG	ACTCACCTCTTCAGAACGAATTGCCA
*CXCL5*	TGCGTTGTGTTTGCTTAACCG	CTTCCACCGTAGGGCACTG
*CXCL10*	CCAAGTGCTGCCGTCATTTTC	GGCTCGCAGGGATGATTTCAA
*GAPDH*	AGGTCGGTGTGAACGGATTTG	TGTAGACCATGTAGTTGAGGTA

## Data Availability

Not applicable.

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
