# Peer review of "Cycloastragenol: A Novel Senolytic Agent That Induces Senescent Cell Apoptosis and Restores Physical Function in TBI-Aged Mice"

_ijms, 2023, doi:10.3390/ijms24076554_

Round 1

Reviewer 1 Report

This is an article focused on the effects of natural compound cycloastragenol (CAG) in preclinical models of DNA damaged cells and mice subjected to irradiation to induce aging. In particular, cellular senescence is considered as the target mechanism for CAG. The report may add novelty in terms of potential senolytic effects that prevent from detrimental effects of aging on physical function.

However, here are some major points to be addressed:

1. CAG has shown a plethora of pharmacological effects and mechanisms only in preclinical models of disease. This should be stated in the introduction.

2. The authors should provide list and data (including assay characteristics) of natural compounds tested in the initial screening.

3. Figure 1. Panel D shows C12FDG staining of IMR-90 cells. Counterstain of live cells should be done in order to distinguish specific ssnolytic effect of CAG.

4. Figure 2. The CAG effect is associated with toxicity through apoptosis. WB showing changes on markers (panel C and D) should be quantified by densitometry.

5. Figure 5. IL-6 blot should be repeated or removed. Again, quantitative analysis should be done for all WB. Panel E: Is CAG effect on cell migration direct? CAG treatment effect on HELF migration should be tested and ANOVA analysis should be carried out. In legend, (F) EF24 should be replaced by CAG.

6. Figure 4. Can the authors determine and show the amount of compound present in the plasma of treated animals? CAG is a molecule with poor water solubility, fast metabolic conversion and low oral bioavailability.

7. Figure 4. Were physical function markers in the tissues (eg skeletal muscle) investigated after the treatment to support the phenotypic findings?

8. Figure 5. A brief discussion compiling in vitro and in vivo data on apoptosis pathway should be included.

Author Response

Dear reviews

Thank you for your efforts in reviewing our manuscript entitled “Cycloastragenol is a novel senolytic agent that induces senescent cell apoptosis and restores physical function in aged mice” (ijms-2202980). We are grateful to the editor for giving us the opportunity to revise our manuscript and judging our work as potentially important and within the scope of International Journal of Molecular Sciences. Accordingly, we have addressed all concerns raised by the reviewers with changes in the form of presentation. We are grateful for these valuable comments and feel that after incorporating the reviewer advice, the revised manuscript has been greatly strengthened, and the significance of our review is now more evident. The detail changes are listed in the attachments.

Author Response

(The authors gave the same response as above.)

Reviewer 3 Report

Zhang et al reported a study on the mechanism of anti-aging effects of cycloastragenol in both in vitro and in vivo models. It is of great interest to the field as it shows a promising way to delay aging by combination use of synergistic compounds and thus reducing toxicity of individual agents. The manuscript is well organized but may be improved in a few aspects of its results.

1. In terms of cellular senescence marker PAI-1 and p21 (Figure 1F and Figure 3D), a non-senescence control (e.g. NCs) is missing. This is important to show that VP-16 treatment/SCs-CM did promote expression of these two markers and thus senescence.

2. For most qPCR and western blot results, GAPDH was used as the loading control. However, it is suggested that expression of GAPDH may change upon treatment, especially in the context of cellular senescence (e.g. PMID: 28699239, PMID: 31562345). Therefore, a second control can help to improve the validity of these results. Total amount of loaded protein samples may also give some clues.

3. Some western blot images are blurring and difficult to evaluate, e.g. Figure 2D, Figure 5B, Figure 5E. Apparently some of them were over exposed. Could the authors provide independent biological replicates and perform a quantitative analysis?

4. For Figure 3B, secreted cytokines in conditioned medium, how did the authors normalize them? As CAG treatment induced evident cell death, are these cytokines secreted from the remaining live cells or those dead cells before they died? Again, a non-senescence control is missing in this panel.

5. Original images of western blot results are missing. Please include them as supplementary figures.

6. In Figure 4B, when did the body weight monitoring start? The figure shows only 66 days on x-axis while the time from TBI to sacrifice is 81 days. There is discrepancy in initial body weight among the three groups and the change in percentage of initial body weight may be more informative.

7. An approved protocol number from an Ethics Committee is required for mouse experiments described in the manuscript.

8. Some of cited reference are not relevant. For example, refs 31 and 42 do not support the comments in line 443~446.

Minor points:

1. As is described in line 206, the authors have screened 10 natural compounds. It would be more informative if a figure to compare these 10 compounds are shown as supplemental.

2. In right panel of Figure 2B, “Radio” should be “Ratio”.

3. It is stated CAG “selectively induces SCs apoptosis but has no cytotoxicity to NCs”, in line 403. This is not guaranteed by the results shown here.

Author Response

(The authors gave the same response as above.)

Round 2

Reviewer 2 Report

The quality of the paper is much better in the actual form.

However, some WB quantifications are still missing (Fig. 2C-D, 3B, 5B), and no error bars are shown in fig 3C. Corrections of the aforementioned points would be greatly appreciated before final publication.

Author Response

Thank you for your efforts in reviewing our manuscript entitled
“Cycloastragenol is a novel senolytic agent that induces senescent cell
apoptosis and restores physical function in aged mice” (ijms-2202980).
We are grateful to the editor for giving us the opportunity to revise our
manuscript and judging our work as potentially important and within the
scope of International Journal of Molecular Sciences. The detail changes
are listed as following.

Reviewer 3 Report

The authors have done a great study. All my points are well answered in your response. I'd like to add some words to the minor point 3. I said "selectively induces SCs apoptosis but has no cytotoxicity to NCs" was not guaranteed, as only cell viability was examined in NCs. This does not sufficiently exclude other cell damages under the term "cytotoxicity". Therefore it would be state something like "selectively induces cell death in SCs but not NCs", instead of "no cytotoxicity". Sorry for the ambiguity.

Author Response

(The authors gave the same response as above.)
